# A defect-resistant Co–Ni superalloy for 3D printing

Sean P. Murray [1], Kira M. Pusch[1], Andrew T. Polonsky [1,2], Chris J. Torbet[1], Gareth G. E. Seward[3], Ning Zhou[4], Stéphane A. J. Forsik [4], Peeyush Nandwana [5], Michael M. Kirka[5], Ryan R. Dehoff[5], William E. Slye[4] & Tresa M. Pollock [1✉]

Additive manufacturing promises a major transformation of the production of high economic value metallic materials, enabling innovative, geometrically complex designs with minimal material waste. The overarching challenge is to design alloys that are compatible with the unique additive processing conditions while maintaining material properties sufficient for the challenging environments encountered in energy, space, and nuclear applications. Here we describe a class of high strength, defect-resistant 3D printable superalloys containing approximately equal parts of Co and Ni along with Al, Cr, Ta and W that possess strengths in excess of 1.1 GPa in as-printed and post-processed forms and tensile ductilities of greater than 13% at room temperature. These alloys are amenable to crack-free 3D printing via electron beam melting (EBM) with preheat as well as selective laser melting (SLM) with limited preheat. Alloy design principles are described along with the structure and properties of EBM and SLM CoNi-base materials.

---

[1] Materials Department, University of California Santa Barbara, Santa Barbara, CA 93106, USA. [2] Materials Mechanics and Tribology Department, Sandia National Laboratories, Albuquerque, NM 87185, USA. [3] Department of Earth Science, University of California Santa Barbara, Santa Barbara, CA 93106, USA. [4] R&D Department, Carpenter Technology Corp., Reading, PA 19612, USA. [5] Materials Science and Technology Division, Oak Ridge National Laboratory, Oak Ridge, TN 37830, USA. ✉email: tresap@ucsb.edu

Metal-based additive manufacturing (AM), or three-dimensional (3D) printing, has enabled the fabrication of near net shape metal components with optimized geometries that are not achievable through conventional manufacturing techniques. The promise of increased design flexibility has led to significant interest in applying 3D printing methods to commercial alloys currently used in biomedical, automotive, and aerospace applications[1–3]. However, only a limited number of existing alloys are amenable to the complex thermal conditions present during metal-based AM, where layer-by-layer growth of the component is achieved through local melting of metal powder by either a laser or electron beam energy source[4–7]. AM of metals is fundamentally a repeated welding process, in which a directed energy source is used to locally melt and join material. Candidate materials for AM therefore tend to be weldable alloys, which are less susceptible to cracking mechanisms that originate in the liquid phase, such as liquation cracking or hot tearing, or due to stresses that develop in the solid state, resulting in, for example, strain-age cracking and ductility-dip cracking[8].

Due to their excellent mechanical properties at elevated temperatures, Ni-base superalloys are the material of choice for structural components such as single crystal (SX) turbine blades and vanes used in the hot sections of aircraft engines and land-based natural gas turbines[9]. These alloys consist of a high volume fraction (>0.6) of sub-micron size cuboidal precipitates of the $\gamma'$ phase (Ni$_3$(Al,Ti), L1$_2$) that are coherent with a solid solution-strengthened matrix, or $\gamma$ phase (Ni, A1). However, many of the highest-performing Ni-base superalloys are observed to be non-weldable due to the rapid precipitation of the $\gamma'$ phase shortly after solidification, which impedes the relaxation of thermal stresses by strengthening the recently solidified material, resulting in strain-age cracking[10]. This behavior is captured in Prager–Shira weldability diagrams that show how increasing Al or Ti content, which both increase the volume fraction of the $\gamma'$ strengthening phase, is a reliable proxy for decreasing alloy weldability[11].

As the $\gamma$ phase solidifies, the liquid becomes locally enriched by the rejection of $\gamma'$ forming elements such as Al, Ti, and Ta[12]. This solute segregation lowers the local liquidus temperature, creating solute-enriched liquid films between solid dendrites that contract at differential rates in the melt pool during cooling, resulting in tensile stresses and cracking[13–15]. This susceptibility to liquid-mediated cracking can be influenced by control of the liquid composition and liquid fraction at a given temperature via changes in the alloy composition. Just below the melting point, stresses can be accommodated by plastic deformation in the solid state, which is very sensitive to the temperature at which the strengthening precipitates appear. Therefore, many of the desirable high $\gamma'$ volume fraction Ni-base superalloys, which have a narrow temperature window from the point at which the material solidifies to the temperature at which the precipitates become thermodynamically stable, are susceptible to both cracking in the nearly solidified state and in the solid state. The solute segregation and precipitation processes can, in principle, be modified by global changes in composition.

The cracking susceptibility of high-performance engineering alloys including high $\gamma'$ volume fraction Ni-base superalloys, high strength aluminum alloys and refractory alloys[8,16–20] represents the major barrier to the use of these alloys for AM components in critical applications. For alloys that operate at relatively low temperature, such as high strength aluminum alloys, control of grain nucleation in the melt pool via functionalization of powder surfaces can mitigate the cracking problem[21]. However, this results in a small grain size, which is unfavorable for high-temperature operation. Thus, innovative alloy designs are needed for AM, especially for more severe environments[22].

Several different strategies have recently been pursued for the development of alloys for AM[23]. By increasing the solid solution strengthening elements in the Ni-base superalloy Hastelloy X within the existing commercial alloy composition range, a reduction in microcracking in AM was observed[24]. To control material anisotropy, Haines et al. performed a sensitivity analysis with a focus on adjusting alloy composition to control the columnar to equiaxed transition in Ni-base alloys[25]. Similarly, control of the columnar to equiaxed transition through AM process control has been employed by Kontis et al. to successfully fabricate a non-weldable Ni-base superalloy through atomic-scale grain boundary engineering[26]. Additionally, AM allows for the mixing of alloy powders before printing, resulting in the fabrication of metal–metal composites with unique microstructures that would be difficult to fabricate by other means[27]. Since the $\gamma$-$\gamma'$ microstructure present in modern Ni-base superalloys is desirable due to the excellent mechanical properties it confers, we seek to design an alloy that contains a high $\gamma'$ volume fraction while retaining good printability.

Recent interest in Co-base superalloys commenced with the study of Sato et al. that revealed the possibility of precipitation strengthening in the Co–Al–W ternary system[28]. These Co-base alloys are morphologically identical to their Ni-base counterparts, except the $\gamma'$ strengthening phase is based on Co$_3$(Al,W). The $\gamma'$-strengthened Co-base superalloys have recently been fabricated as SXs via Bridgman growth and polycrystals via wrought processing[29–32].

Here we present a CoNi-base superalloy that can be processed through both selective laser melting (SLM) and electron beam melting (EBM) manufacturing pathways, resulting in crack-free components in spite of the presence of a high volume fraction of the desirable $\gamma'$ phase. A low degree of solute segregation during solidification reduces the susceptibility for liquid-mediated cracking, and the reduced $\gamma'$-solvus temperature alleviates cracking once solidification is complete. Room temperature tensile testing reveals that CoNi-base superalloys have an excellent combination of ductility and strength compared to other Ni-base superalloys currently being investigated for AM. Our approach demonstrates that the CoNi-base superalloy compositional space provides opportunities for the development of superalloys that can leverage the potential of AM.

## Results

**Design approach.** The primary goals for alloy design were good high-temperature strength, achieved via stabilization of a high volume fraction of the $\gamma'$ phase to at least 1100 °C, solidus and liquidus temperatures above 1300 °C with a narrow equilibrium freezing range, high resistance to oxidation provided by formation of alumina in oxidizing environments and favorable printability. Ni-base alloys strengthened with high volume fractions of $\gamma'$ phase are well known for their tendency to crack during powder-bed fusion (PBF) printing[33]. In these alloys the Ni$_3$Al phase is thermodynamically stable up to a few degrees below the solidus temperature, thus it was hypothesized that a gap of >50 °C between the solidus and solvus temperatures would increase cracking resistance while still maintaining a high volume fraction of precipitates at elevated temperatures.

The emergence of this class of cobalt-base alloys coincides with a recent thrust to develop computational and high-throughput alloy design tools[34]. This has enabled a suite of tools to be developed and integrated for exploration of the large multi-component composition-space necessary to discover alloys for AM. These tools were used to expand compositions from the original Co–Al–W ternary to the multicomponent composition of

the printed alloy investigated here. Below are brief descriptions of the tools and their role in the alloy design process.

The strength of the two-phase $\gamma - \gamma'$ microstructure is governed by the resistance of the L1$_2$ $\gamma'$ phase to shearing by ordinary or partial dislocations entering from the $\gamma$, creating either superlattice intrinsic stacking faults (SISFs) or antiphase boundaries (APBs). To assess the role of solute on these planar fault energies, density functional theory (DFT) calculations using the Vienna ab initio simulation package (VASP) along with special quasi-random structures (SQSs), the axial next-nearest neighbor Ising (ANNNI) model[35,36], and the diffuse multi-layer fault (DMLF) model with a proximate structure for the (111) APB[37,38] were conducted. Calculations indicated that Ti, Ta, Nb, and Ni were favorable alloying additions. Creep experiments on several SX variants of these materials confirmed that alloys with these elements possessed creep strengths that exceeded first generation SX Ni-base superalloy levels[39,40].

To validate fault energy assessments and understand the complex precipitate shearing mechanisms observed across individual and groups of precipitates following mechanical testing, ab-initio calculations of the generalized-stacking-fault (GSF) potential energy were used for phase field dislocation calculations[41,42].

Interfacing with first-principles electronic structure codes, a clusters approach to statistical mechanics (CASM) software package automates the construction and parameterization of effective Hamiltonians and subsequently builds highly optimized kinetic Monte Carlo codes to predict finite-temperature thermo-dynamic and kinetic properties[43,44]. CASM includes cluster expansions for configurational disorder in multi-component solids and lattice-dynamical effective Hamiltonians for vibrational degrees of freedom involved in structural phase transitions. CASM was used to assess the intrinsic thermodynamic stability of the L1$_2$ phase at elevated temperatures, to calculate the ternary Co–Al–W phase diagram and to assess the finite temperature fault energies and the driving forces for segregation at planar faults[45–47].

An early thermodynamic database in the compositional space adjacent to the Co$_3$(Al,W) phase was developed and subsequently incorporated into the CompuTherm PanCobalt database[48]. Thermodynamic calculations were used to adjust the Ni content to increase the L1$_2$ solvus temperature to the desired range, to predict phases present across composition space in the combinatorial studies described below and to predict solidification paths and their likely influence on printability.

Ternary Co–Al–W alloys have limited oxidation resistance due to the formation of non-protective CoO and mixed spinels[49,50]. Because first principles modeling of non-stoichiometric oxides is extremely challenging, a combinatorial library approach was developed. Ion plasma deposition (IPD) using five cathodes of different alloy compositions was used to create three libraries covering a spectrum of Co–Ni–Al–W–Cr–Ta alloy compositions with 234 samples. The combinatorial approach[51,52] was coupled with rapid oxide screening based on photo-stimulated lumines-cence spectroscopy (PSLS)[53] and the Calphad database[48] to outline regions of composition space giving rise to $\alpha$-Al$_2$O$_3$ scales and two-phase $\gamma - \gamma'$ microstructures.

Starting from the composition of an alumina-forming IPD alloy of composition Co–32.4Ni–11.7Al–4.4W–3.3Cr–1.5Ta (at.%), three additional alloys with small variations in composi-tion were arc melted into 40 g buttons[40]. After analysis of microstructure and phases present, a final composition, SB-CoNi-10, possessing a desirable two-phase microstructure was selected for (a) SX growth to measure basic properties and (b) 3D printing to assess its behavior in both laser and electron beam based PBF methods.

The SB-CoNi-10 alloy has a nominal composition of Co–36.4Ni–13.2Al–6Cr–3.5Ta–1W (at.%) along with minor additions of 0.08B–0.08C–0.018Hf–0.002Y which are included for grain boundary strengthening, carbide formation, and oxide scale adhesion[40]. It possesses a high volume fraction of $\gamma'$ of ~0.7 after aging at 1000 °C, a mass density similar to Ni-base superalloys (8.65 g cm$^{-3}$), and is able to form environmentally protective $\alpha$-Al$_2$O$_3$ after high-temperature exposure in air[40,54]. The $\gamma'$-solvus, solidus, and liquidus values for this alloy have been measured by differential thermal analysis as 1204, 1329, and 1381 °C, respectively, resulting in a window of 125 °C where the material is single phase $\gamma$ and an equilibrium freezing range of 52 °C.

**Alloy synthesis**. A 136 kg lot of SB-CoNi-10 powder was fabri-cated by vacuum induction melting and argon gas atomization. A powder size range of 15–53 μm was used for SLM printing, while a larger size distribution of 53–177 μm from the same heat was used for the EBM process, as shown in Fig. 1a and b. Rectangular blocks and blade-shaped samples were printed in both processes with process parameters that are typical of those employed for printing of Ni-base alloys. The laser printing was conducted with a preheat temperature of 200 °C applied to the powder bed, while the EBM printing utilized the electron beam to preheat the powder bed to 1000 °C.

The 3D printing trials with both the EBM and SLM approaches resulted in crack-free blocks and blade-shaped samples in the as-printed state (Fig. 1c–f). Due to the favorable solvus temperature and the thermal conditions present during printing, the block samples were suitable for tensile tests without the need for post-processing heat treatments. Additional tensile tests were performed after subjecting the alloy to a standard processing route for additively manufactured components, which consists of (i) hot isostatic pressing (HIP) to close any residual gas or shrinkage porosity, (ii) solution heat treatment (SHT) above the $\gamma'$-solvus to homogenize the alloy, and (iii) a lower temperature aging to precipitate and coarsen the $\gamma'$ phase. These specimens are denoted as HIP + SHT + aged. Further details on printing and post processing are given in the "Methods" section.

The as-printed grain structure characterized by electron backscatter diffraction (EBSD) in the scanning electron micro-scope (SEM) is shown for both the SLM and EBM processes with inverse pole figure (IPF) maps in Fig. 1g and h. In the higher preheat EBM process, there is a greater tendency for grains to be columnar and to assume the preferred solidification growth direction of ⟨001⟩ aligned with the build direction. There are occasional clusters of equiaxed grains, which are associated with fluctuations in thermal conditions and not due to lack of fusion locally. In the low preheat SLM process, the grains are also columnar but have greater variation in their crystallographic orientation relative to the build direction.

**As-printed chemical segregation**. In order to evaluate solute segregation in the as-printed condition, both EBM and SLM alloys were investigated by electron probe microanalysis (EPMA) along the XY-plane, i.e. perpendicular to the build direction, from cross-sections taken 1 mm below the final build layer (Fig. 2). By collecting a 20 × 20 point grid of compositional data over a 100 × 100 μm area in the center of the regions shown in Fig. 2b and c, statistics on the alloy composition at various stages of the soli-dification process were collected. These measurements have been repeated on a version of SB-CoNi-10 that was fabricated into a SX by the Bridgman method using a 1 mm × 1 mm area grid for data collection over the dendrite florets shown in Fig. 2a. Accurate compositional measurement requires that the electron probe (and

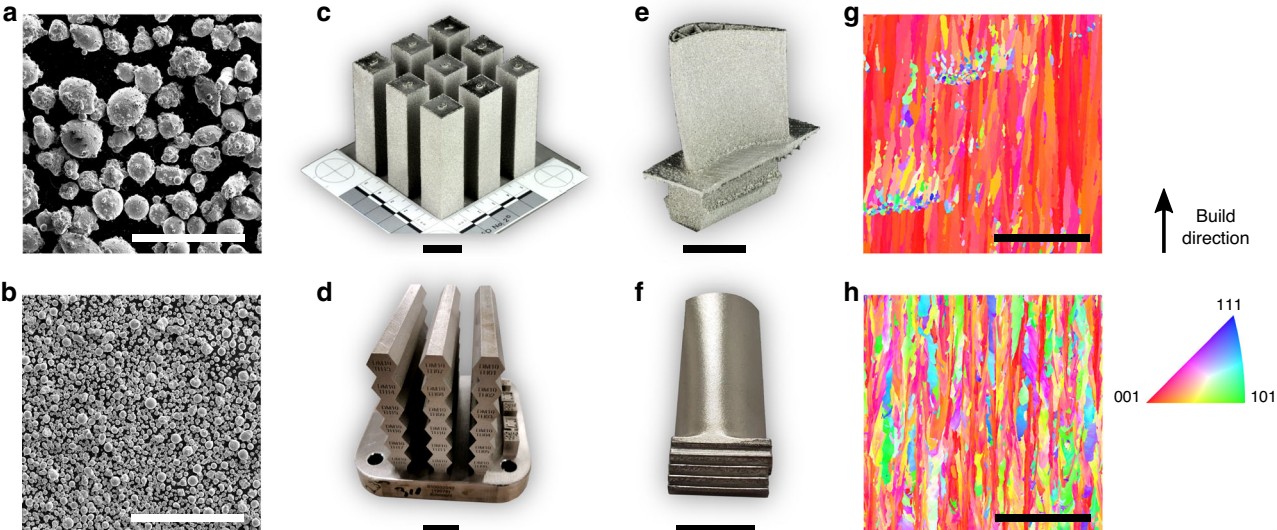

**Fig. 1 Additive manufacturing of a CoNi-base superalloy through EBM and SLM.** SEM micrographs of metal powder of SB-CoNi-10 used for **a** EBM and **b** SLM printing trials. Simple bar geometries have been printed for uniaxial tensile testing **c**, **d** in addition to complex geometries such as prototype turbine blades with **e** internal cooling channels or **f** thin, over-hanging platforms. IPF maps acquired through EBSD show the grain structure of the as-printed CoNi-base superalloy along the build direction manufactured through **g** EBM and **h** SLM. The scale bars for **a**, **b** and **g**, **h** are 500 μm. The scale bars for **c**–**f** are 2 cm.

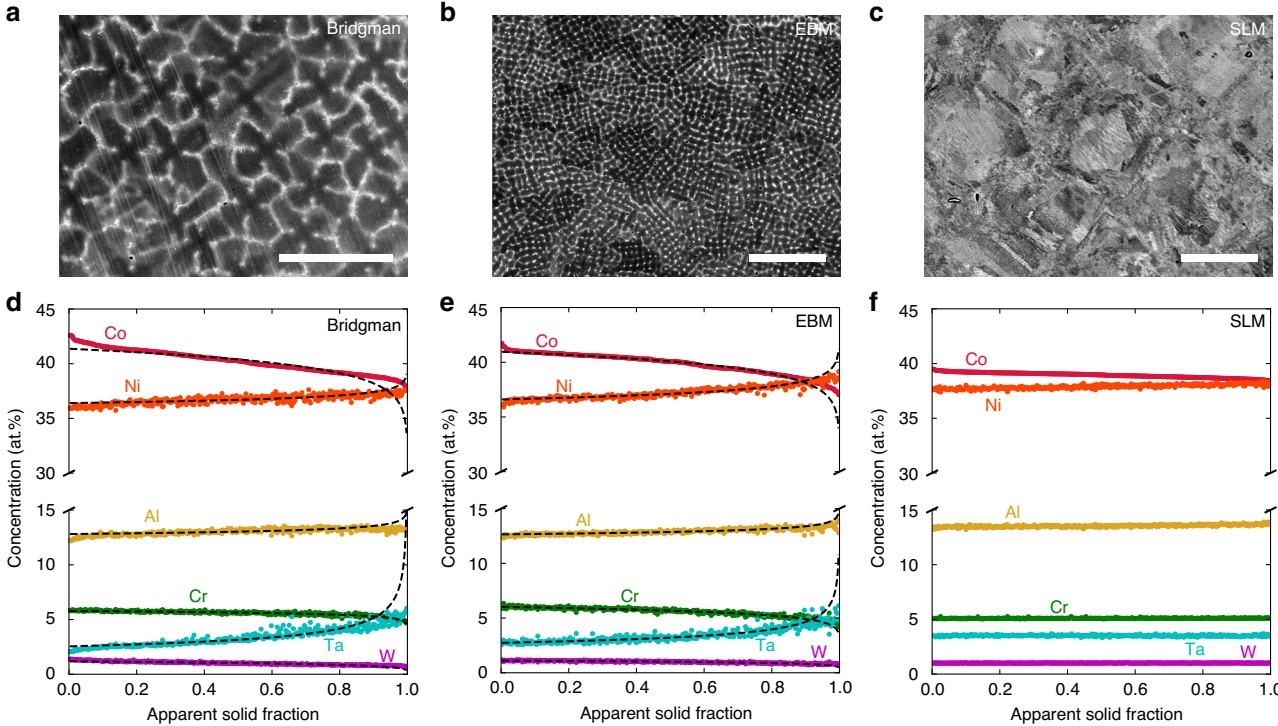

**Fig. 2 As-printed chemical segregation after Bridgman casting, EBM, and SLM.** BSE micrographs of the *XY*-plane microstructures of SB-CoNi-10 after fabrication through **a** Bridgman casting, **b** EBM, and **c** SLM. Quantitative compositional data and Scheil curve fits for the apparent distribution coefficients are shown for the **d** Bridgman, **e** EBM, and **f** SLM samples. EPMA grid scans of 20 × 20 evenly spaced points were collected in the centers of the BSE images shown in **a**–**c** with grid dimensions of **a** 1 × 1 mm and **b**, **c** 100 × 100 μm. Scale bars for **a**, **b**, and **c** are 500, 50, and 50 μm, respectively.

subsequently detected x-rays) interacts with a volume of sample that is homogeneous in composition. The interaction volume of the electron probe with the alloy surface is non-negligible at the accelerating voltages necessary for sufficient characteristic x-ray signal from elements such as W or Ta. Therefore, compositional measurements were taken on the *XY*-plane for all specimens to reduce the impact of compositional fluctuations beneath the alloy surface on the measured values. By sorting this compositional data from highest Co content to lowest Co content, the composition of the solid at each apparent solid fraction, or $f_s$, for each of the major alloying additions can be estimated and is shown in Fig. 2d–f. By fitting these curves with the Scheil equation[55], which describes solute segregation during solidification under the assumption of no diffusion in the solid and infinite diffusion in

**Table 1 Experimentally measured distribution coefficients ($k_i$) using the EPMA grid technique.**

| Alloy name | Alloy class | $k_{Co}$ | $k_{Al}$ | $k_W$ | $k_{Ta}$ | $k_{Cr}$ | $k_{Ni}$ | $k_{Re}$ |
|---|---|---|---|---|---|---|---|---|
| 11W[29] | Co-, SX | 1.01 | 0.94 | 0.99 | – | – | – | – |
| 2Ta[29] | Co-, SX | 1.01 | 0.95 | 1.02 | 0.63 | – | – | – |
| CrTa[29] | Co-, SX | 1.01 | 0.96 | 1.01 | 0.61 | 0.99 | – | – |
| SB-CoNi-10 | CoNi-, EBM | 1.03 | 0.98 | 1.14 | 0.77 | 1.09 | 0.98 | – |
| SB-CoNi-10[40] | CoNi-, SX | 1.03 | 0.97 | 1.20 | 0.69 | 1.04 | 0.99 | – |
| CMSX-4[60] | Ni-, SX | 1.06 | 0.91 | 1.29 | 0.76 | 1.05 | 0.97 | 1.47 |
| SX-series[61] | Ni-, SX | 1.05–1.10 | 0.81–0.89 | 1.52–1.54 | 0.69–0.80 | 1.07–1.17 | 0.94–0.95 | 1.38–1.60 |
| SX-series+C[61] | Ni-, SX | 1.03–1.13 | 0.86–0.90 | 1.36–1.44 | 0.76–0.89 | 1.05–1.13 | 0.93–0.96 | 1.33–1.49 |

the liquid, the apparent distribution coefficients that describe the intensity of solute segregation, $k$, can be experimentally determined. The distribution coefficient $k$ is defined as $k = C_s/C_l$ where $C_s$ and $C_l$ are the composition of the solid and liquid on either side of the solid–liquid interface during solidification. These results are shown in Table 1. Apparent distribution coefficients measured in Bridgman-grown SXs of SB-CoNi-10 show the same partitioning tendencies as the EBM sample[40]. The degree of partitioning in SB-CoNi-10 is much lower in comparison to conventional Ni-base alloys, resulting in the variation of the fraction liquid with temperature also being lower[40]. Since the length-scale of the segregation present in the as-printed cellular structure is smaller than the interaction volume of the EPMA electron probe, it is not possible to resolve the apparent distribution coefficients for the SLM sample. This is demonstrated by the flat Scheil curves in Fig. 2f where Scheil curve fits are not applied in this case.

Quantitative chemical maps of the EBM and SLM material are provided in Fig. 3a and b for visualization of this solute segregation, showing that Co, Cr, and W segregate to the dendrite cores and that Ni, Al, and Ta segregate to the interdendritic regions. As noted above, the fine-scale segregation in the SLM samples cannot be resolved with this technique.

**Microstructure evolution**. The high preheat temperature employed in EBM (1000 °C) reduces the thermal stresses that develop during solidification, with low orientation gradients present within individual grains (Figs. 1g and 4f). In the case of precipitation strengthened alloys such as SB-CoNi-10, preheating results in $\gamma'$ microstructural evolution that varies with the build height[56]. Investigations at varying heights in the EBM build (Fig. 4b–e) reveal that near the top of the build the $\gamma'$ phase precipitates are at a length-scale that is difficult to resolve by scanning electron microscopy, and that the $\gamma'$ phase coarsens along the depth of the build. Traditional post-processing can be applied to this alloy in order to homogenize the chemical segregation and eliminate gradients in precipitate size. HIP applied above the $\gamma'$-solvus temperature ($T_{HIP} = 1245$ °C) resulted in a microstructure that was free from porosity and also resulted in recrystallization of the microstructure into a coarse-grained equiaxed microstructure (Fig. 4g). While the $\gamma'$ phase has a non-uniform size distribution along with non-cuboidal precipitate shapes after the HIP, a short 2 h SHT at 1245 °C followed by aging at 1000 °C for 50 h produces the $\gamma'$ morphology shown in Fig. 4h and i with a measured volume fraction of ~0.7. This heat treatment schedule has also been applied to SXs of CoNi-alloys grown by the Bridgman method, and could be modified if changes in the grain structure or precipitate sub-structure are desirable[40].

The limited preheat used during SLM (200 °C) does not promote the formation and coarsening of the $\gamma'$ phase during printing as occurs in EBM. The prior weld pools are visible in Fig. 5a with columnar grains that grow along the build direction from the bottom of the weld pools and grains that grow laterally from the walls of the weld pools towards the laser track centerline, along with limited amounts of porosity. BSE micrographs at various depths below the final build layer shown in Fig. 5b–e reveal that the cellular microstructure persists throughout the build. Additionally, where the EPMA mapping in Fig. 3b does not detect chemical segregation, these micrographs reveal that chemical segregation does persist at the cellular level along with the formation of bright Ta-rich carbides (Fig. 5i). A subsolvus HIP followed by a supersolvus SHT resulted in recrystallization of the as-printed microstructure (Fig. 5f and g) and aging produced the $\gamma'$ morphology shown in Fig. 5h and i.

**Mechanical testing of as-printed and post-processed alloys**. Room temperature quasi-static tensile tests were performed on both the SLM and EBM printed material. The specimens were machined from bars such as the ones presented in Fig. 1c and d and tests were performed on both the as-printed and HIP + SHT + aged microstructures. A summary of the mechanical strength results is shown in Table 2[33,57]. Samples tested with the tensile axis along the build direction are indicated as Z-orientation, whereas samples that were tested perpendicular to the build direction are indicated as XY-orientation. Typically, additively manufactured alloys that contain a high density of cracks after printing fail shortly after the yield point. The SB-CoNi-10 alloy demonstrates significantly improved ductility and resistance to cracking compared to similar Ni-base superalloys produced by PBF, as shown by the engineering stress–strain curves in Fig. 6a and f. SEM investigations of the fracture surfaces after testing revealed that failure was ductile and intragranular in all room temperature tests (Fig. 6b–e, g–j). This is evidenced by the tendril-like features that are indicative of microvoid coalescence, along with the significant reductions in cross-sectional area of all specimens after testing. Severe elliptical distortions in the fracture surfaces of the as-printed specimens are visible in Fig. 6b and g. This is a result of rotations of the columnar grains where the slip planes of the dominant slip system reorient in order to maximize the resolved shear stress they experience[58,59]. EBSD maps along the build direction of the post-mortem EBM specimens reveals this grain rotation by showing how the ⟨001⟩ crystallographic direction of the columnar grains are now more strongly aligned with the loading direction after the test (Figs. 4f and 7a). Additionally, these EBSD maps show how strain accumulates in grains oriented favorably for dislocation slip in the HIP + SHT + aged samples (Fig. 7c and d) and that the post-test microstructures of both specimens are free from cracking along grain boundaries (Fig. 7a–d).

**Discussion**
By utilizing a suite of modern research tools, including first-principles calculations[35,36,43,44,51], high-throughput thermodynamic database calculations[40,48], combinatorial alloy

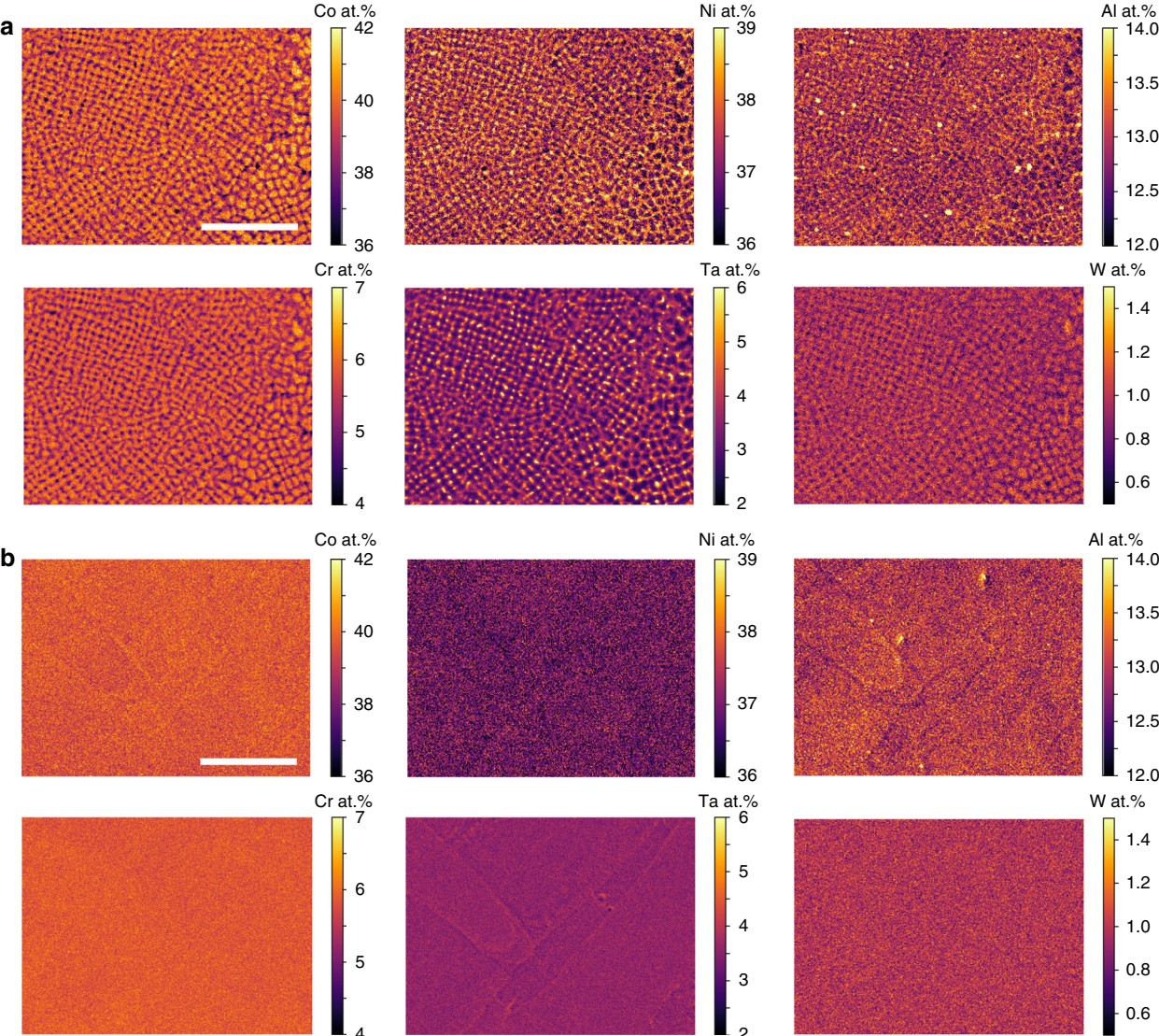

**Fig. 3 Quantitative EPMA maps.** EPMA elemental composition maps of the *XY*-plane microstructures for the **a** EBM and **b** SLM samples. Co, Cr, and W segregate to the dendrite core while Ni, Al, and Ta segregate to the interdendritic regions. Each map has a step size of 0.5 μm. The scale bars for **a** and **b** are 100 μm.

synthesis[51,52] and non-destructive characterization techniques[53], we have rapidly explored a complex multicomponent space for Co–Ni alloy compositions that are favorable for AM and also possess favorable mechanical and environmental properties. A particular challenge for alloy design in the additive domain is the need to fabricate expensive powders in significant quantities (typically >100 kg); thus the computational suite of design tools was essential for efficient exploration and selection of a composition likely to be favorable for AM. By fabricating the alloy SB-CoNi-10 with both SLM and EBM methods, it is suggested that this region of compositional space may yield alloys that are easier to process through AM compared to current Ni-base superalloys. Importantly, it has been demonstrated that the SB-CoNi-10 alloy can be printed in a simple bar form (Fig. 1c and d), which is subject to less mechanical constraint during printing, and in a blade-shaped form (Fig. 1e and f), which would be more prone to cracking due the added thermomechanical gradients associated with this geometry.

The EPMA grid technique described above is often used on SX castings of superalloys fabricated using the Bridgman method to study the microsegregation of various elements during directional

solidification, which strongly affects the liquid density during solidification and can promote casting defects such as stray grains and freckle chains[29,60,61]. The solidification segregation in the SB-CoNi-10 alloy and related Co-base alloys is much less pronounced compared to Ni-base alloys (Table 1) with distribution coefficients near 1 for all elements except for Ta which is ~0.75. The less pronounced segregation in the Co-rich alloys, particularly in the late stages of solidification, is beneficial for cracking resistance[14,15]. The intensity of the segregation is similar in the EBM material compared to the SX material, though the scale of the dendritic structure is clearly refined in the EBM material due to the higher cooling rates in the EBM process[62]. Similar to SX growth processes which exhibit an annealing effect due to slow withdrawal rates, the layer-by-layer thermal excursions in the EBM process might be expected to influence the observed solute segregation due to solid-state diffusion. However, prior research on segregation in these materials has shown these effects to be minor[29]. In the SLM process, with even higher thermal gradients and cooling rates (>10$^6$ K/m at the solid–liquid interface[63]), the structure is refined further still. Finer scale chemical maps acquired by transmission electron microscopy are required to

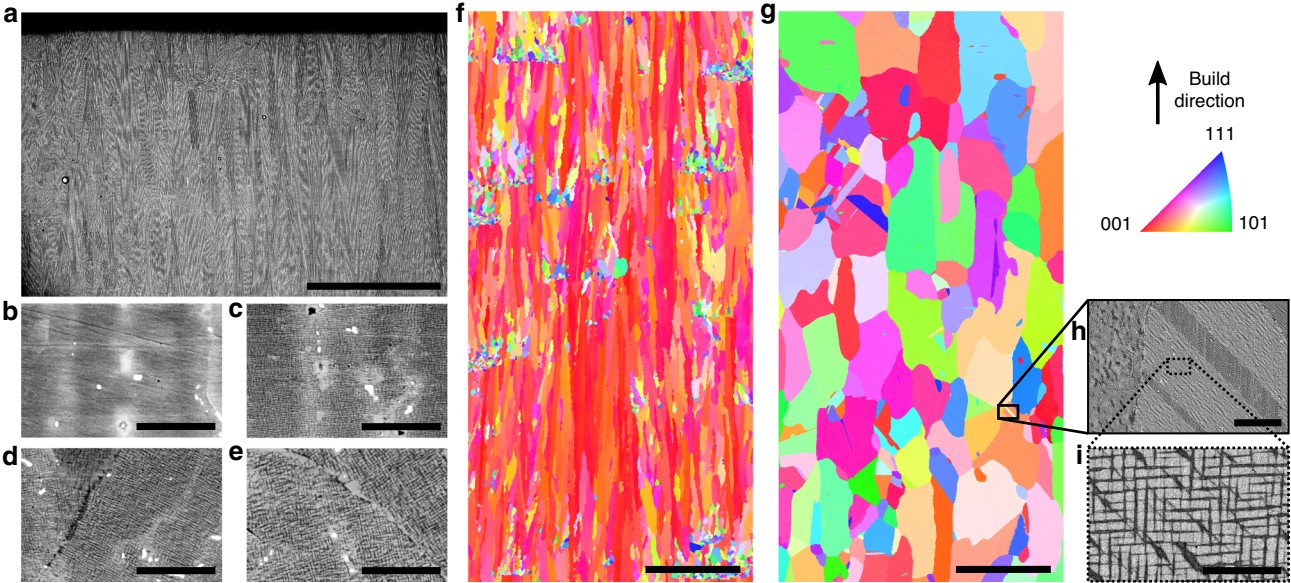

**Fig. 4 EBM microstructural evolution before and after post-processing. a** Stitched BSE image of the final build layers in as-printed EBM SB-CoNi-10. BSE micrographs of the as-printed EBM alloy at different depths below the final build layer: **b** near the final build layer, **c** 1 mm below, **d** 2 mm below, and **e** 4 mm below. **f** IPF map of the as-printed EBM alloy and **g** IPF map of the HIP + SHT + aged material. Both EBSD scans were acquired at a similar distance from the final build layer (~22 mm) that is representative of the center of the tensile specimen gauge sections. **h**, **i** Additional BSE micrographs of the $\gamma - \gamma'$ microstructure after post-processing. The scale bar for **a** is 500 μm. The scale bars for **b**–**e** are 5 μm. The scale bars for **f** and **g** are 500 μm. The scale bar for **h** is 25 μm. The scale bar for **i** is 5 μm.

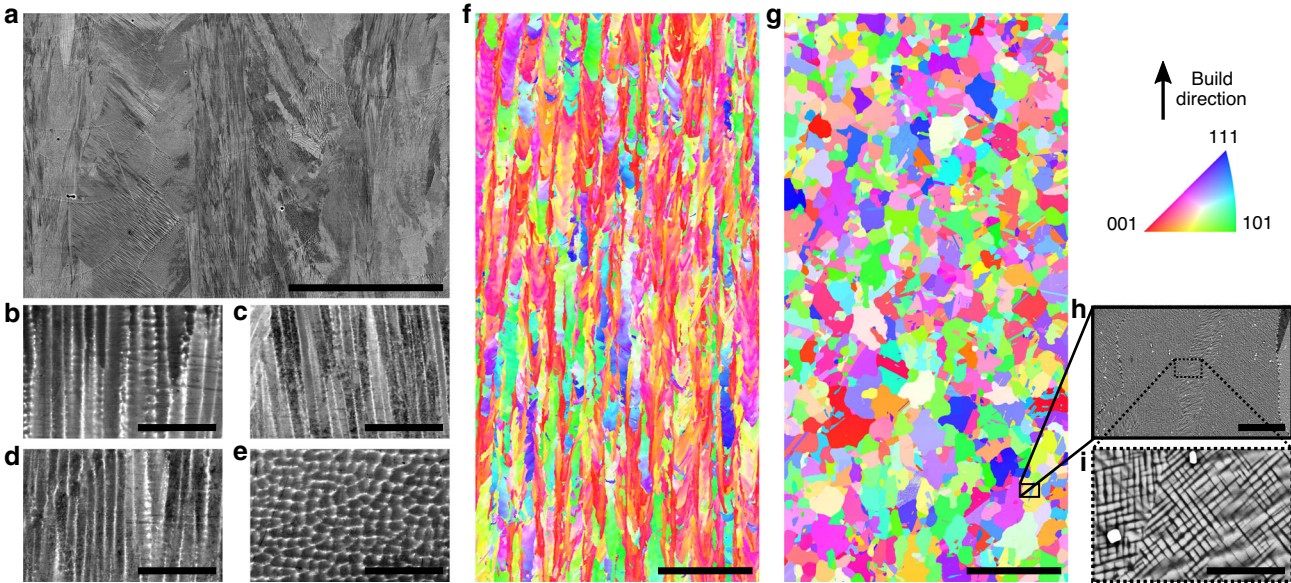

**Fig. 5 SLM microstructural evolution before and after post-processing. a** Stitched BSE image of the as-printed SLM microstructure of SB-CoNi-10 with characteristic melt pool boundaries visible. BSE micrographs of the as-printed EBM alloy at different depths below the final build layer: **b** near the final build layer, **c** 1 mm below, **d** 2 mm below, and **e** 4 mm below. **f** IPF map of the as-printed SLM alloy and **g** IPF map of the HIP + SHT + aged material. **h**, **i** Additional BSE micrographs of the $\gamma - \gamma'$ microstructure after post-processing. The bright particles are Ta-rich carbides which are a result of the addition of carbon. The scale bar for **a** is 50 μm. The scale bars for **b**–**e** are 5 μm. The scale bars for **f** and **g** are 500 μm. The scale bar for **h** is 25 μm. The scale bar for **i** is 5 μm.

conclusively measure the degree, if any, to which segregation is suppressed in the SLM processed samples.

The alloys described above are readily modified through standard post-processing such as HIP and heat treatment to remove microstructural inhomogeneities present after printing, such as chemical segregation and porosity, and to promote a more equiaxed grain structure after recrystallization. In the case of EBM, where long columnar grains aligned with ⟨001⟩ along the build direction are observed in the as-printed state, modified heat treatments may be desirable to preserve the as-printed grain structure while removing as-printed segregation and pores, as has been demonsrated in EBM processed CMSX-4[64]. The microstructure after EBM resembles the columnar grain structure formed in directionally solidified (DS) castings used in the aerospace industry for superalloy turbine blades. The elimination of transverse grain boundaries in DS castings

**Table 2 Results of room temperature tensile testing.**

| Alloy | Heat treatment | Orientation | $\sigma_y$ (MPa) | UTS (MPa) | Elongation (%) |
|---|---|---|---|---|---|
| EBM SB-CoNi-10 | As-printed | Z | 593 | 1281 | 33.2 |
| EBM SB-CoNi-10 | HIP + SHT + aged | Z | 525 | 1185 | 35.7 |
| EBM SB-CoNi-10 | HIP + SHT + aged | Z | 518 | 1183 | 32.0 |
| EBM CM 247®[33] | As-printed | Z | 894 | 1196 | 9.8 |
| EBM CM 247®[33] | HIP + SHT + aged | Z | 825 | 1102 | 9.2 |
| EBM CM 247®[33] | HIP + SHT + aged | Z | 838 | 853 | 1.2 |
| SLM SB-CoNi-10 | As-printed | XY | 753 | 1289 | 13.1 |
| SLM SB-CoNi-10 | HIP + SHT + aged | XY | 689 | 1379 | 23.2 |
| SLM SB-CoNi-10 | As-printed | Z | 717 | 1110 | 24.8 |
| SLM SB-CoNi-10 | HIP + SHT + aged | Z | 621 | 1289 | 23.3 |
| SLM IN 738LC[57] | As-printed | XY | 895 | 1010 | 1.6 |
| SLM IN 738LC[57] | HIP + SHT + aged | XY | 926 | 926 | 0.8 |

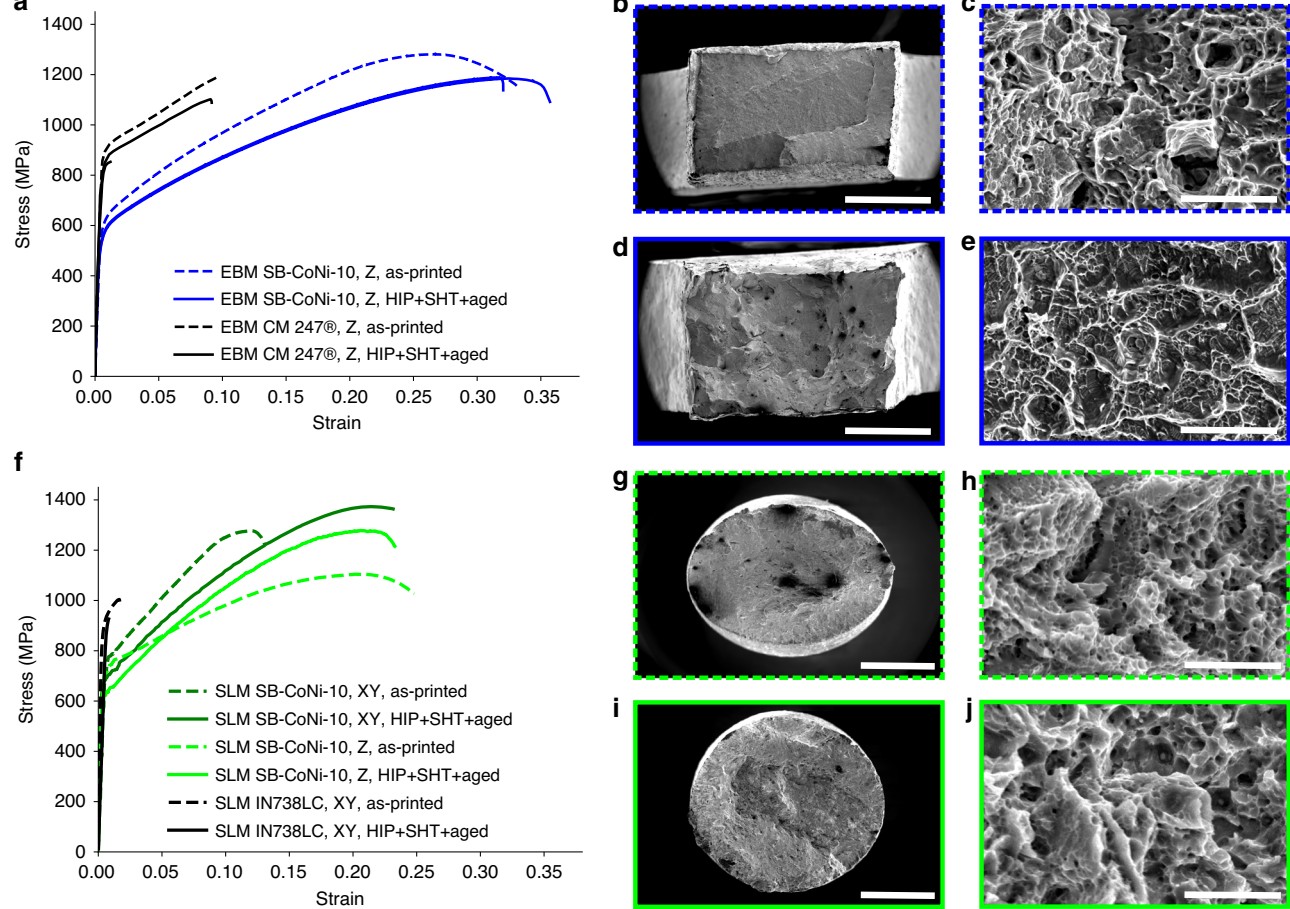

**Fig. 6 Tensile testing of EBM and SLM SB-CoNi-10 at room temperature.** Stress–strain curves for quasi-static tensile tests at room temperature on the **a** EBM and **f** SLM materials in the as-printed and HIP + SHT + aged conditions compared to EBM CM 247[33] and SLM IN738LC[57]. SEM fractography of the **b**–**e** EBM samples and the **g**–**j** SLM samples in the **b**, **c**, **g**, **h** as-printed and **d**, **e**, **i**, **j** HIP + SHT + aged conditions reveal features indicative of ductile fracture in all specimens. The higher magnification images are taken near the center of each fracture surface. The scale bars for **b**, **d** are 1 mm. The scale bars for **g**, **i** are 2 mm. The scale bars for **c**, **e**, **h**, **j** are 5 µm.

significantly improves the high temperature creep response compared to equiaxed castings, which extends the usable life of alloys employed where a combination of high temperature and stress are present. Since these columnar grain structures are industrially relevant, this motivated the printing of the limited amount of EBM powder available for this study in the Z-orientation for microstructural investigations and mechanical testing.

The mechanical response at room temperature demonstrates the excellent printability of the SB-CoNi-10 alloy. The ultimate tensile strength (UTS) and elongation at failure was high with both Z samples with the tensile axis oriented along the build direction (EBM, SLM) and XY samples with the tensile axis oriented normal to the build direction (SLM), demonstrating good properties both parallel and transverse to the axis of the columnar microstructure. In spite of their high UTS after

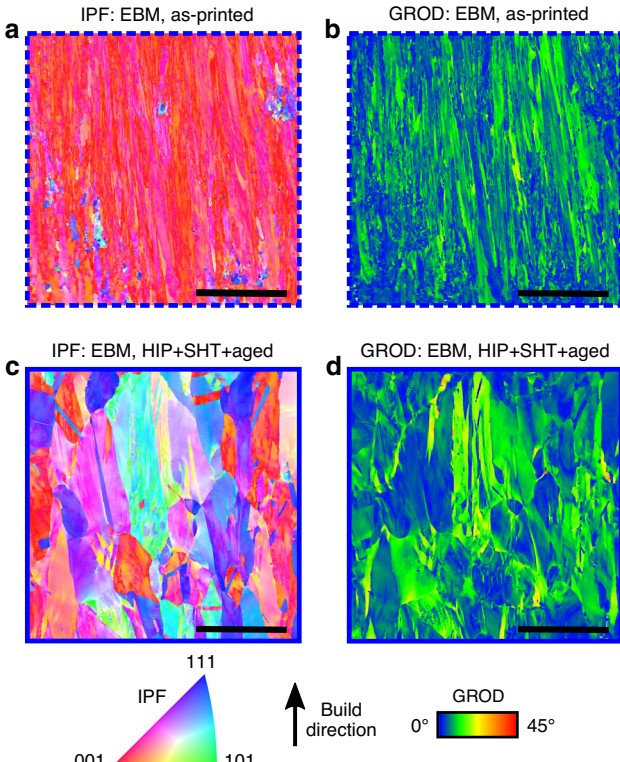

**Fig. 7 EBSD of post-mortem EBM tensile specimens. a, c** IPF maps and **b, d** grain reference orientation deviation (GROD) maps show the accumulation of plastic strain after tensile testing of the EBM material in the **a, b** as-printed and **c, d** HIP + SHT + aged conditions. The scale bars are 500 μm.

significant strain-hardening during plastic flow, tensile ductilities in excess of 32% are observed for EBM SB-CoNi-10, which is higher compared to the standard precipitation strengthened Ni-base alloy CM 247® (CM 247® is a registered trademark of Cannon-Muskegon Corporation) fabricated by EBM[33]. For the SLM SB-CoNi-10 specimens, tensile ductilities in excess of 13% are observed, which is again greater than the standard precipitation strengthened Ni-base alloy IN738LC manufactured by SLM[57]. An anisotropy in the tensile properties is apparent in the SLM samples that depends on build orientation. The as-printed *XY*-oriented samples exhibit higher strength and lower ductility than their *Z*-oriented counterparts, which has been observed in other additively manufactured superalloys with columnar microstructures[65]. This behavior is often reversed when a high density of cracks is present in the as-printed microstructure, resulting in reduced yield strengths as the print orientation changes and preexisting cracks become oriented perpendicular to the loading direction[66]. The SX Ni-base superalloy CMSX-4 has recently been printed in cylindrical bar form by EBM, producing a SX core with equiaxed grains at the surface[67]. Removing samples from the as-printed SX core yielded a room temperature yield strength of 829 MPa and elongation of 6.1%, with no apparent cracking. However, it is unclear whether the techniques used to create the single crystalline core could be used to print this alloy into more complex blade shapes. While the SB-CoNi-10 alloy has a lower yield strength compared to CMSX-4, the CoNi-base superalloy compositional space is under-explored and it is likely that future alloy compositions could be designed with higher yield strengths.

In conclusion, a recently developed CoNi-base superalloy, SB-CoNi-10, has been successfully printed using both EBM and SLM.

Compositional mapping of the as-printed microstructures reveals that favorable solute partitioning combined with favorable $\gamma'$-solvus temperatures results in the suppression of cracking across the range of solidification conditions encountered in the EBM and SLM processes. The high thermal gradients and cooling rates in the printing process results in substantial refinement of the as-solidified structure compared to conventional processing routes, reducing the necessary time for SHT. The alloys are processable through standard post-processing and heat treatments where a fine dispersion of a high volume fraction of the $\gamma'$ phase is precipitated. Tensile testing reveals that these alloys exhibit excellent ductility and a high UTS due to a low propensity for defect formation during printing compared to other high $\gamma'$ volume fraction Ni-base superalloys fabricated by EBM and SLM. This study suggests that further investigation of the CoNi-base superalloy compositional space will be promising for future AM applications. Emerging high throughput experimental and computational tools now enable rapid exploration of the high-dimensional composition-spaces needed to discover alloys for AM.

## Methods

**3D printing parameters and heat treatment.** Metal powder of SB-CoNi-10 for the AM trials was produced in a 136 kg batch by vacuum induction melting and argon gas atomization. Powder was separated into size ranges of −53/+15 μm for SLM and −177/+53 μm for EBM. The EBM fabricated samples were produced on an Arcam Q10+ system running 4.2.89 EBM control software under a controlled He vacuum with an applied preheat temperature of 1000 °C. Other EBM printing details include: max current = 18 mA, speed function = 63 (unitless), hatch spacing = 125 μm, layer thickness = 75 μm, using a standard Arcam raster scan strategy. The SLM fabricated samples were produced on a SLM Solutions SLM125 with an applied preheat of 200 °C in an inert gas atmosphere. Other SLM printing details include: layer thickness = 30 μm, laser power = 134 W/176 W, hatch spacing = 80 μm, laser speed = 744 mm s$^{-1}$/1137 mm s$^{-1}$, using a bidirectional raster scan strategy. The EBM specimens underwent HIP at 1245 °C for 4 h under 103.4 MPa of isostatic pressure in Ar, followed by a SHT at 1245 °C for 2 h and aging at 1000 °C for 50 h under vacuum. The EBM fabricated samples were cooled by furnace shut-off quenching after each heat treatment step. The SLM specimens underwent HIP at 1177 °C for 4 h under 103.4 MPa of isostatic pressure in Ar, followed by a SHT at 1245 °C for 1 h and aging at 1000 °C for 50 h. In the case of the SLM samples, these samples were cooled by an oil quench after the SHT, followed by a furnace quench after the aging heat treatment.

**Microstructural characterization.** Samples were prepared for microscopy using standard metallographic techniques consisting of grinding with SiC papers down to 1200 grit followed by polishing down to a 50 nm colloidal alumina suspension using a Vibromet polisher for 4 h. Scanning electron microscopy with secondary electron (SE) imaging and backscattered electron (BSE) imaging was performed on a ThermoFisher Apreo C at accelerating voltages between 5 and 20 kV using a Schottky field emission gun. EBSD maps were acquired using an EDAX Velocity EBSD camera in a FEI Versa3D microscope at an accelerating voltage of 30 kV. All IPF maps are defined such that the ⟨001⟩ is aligned with the build direction. The collected diffraction patterns were indexed by spherical indexing using the EMSphInx v0.2 software package[68] and Hough indexing within OIM Analysis™ v8 software. Fracture surfaces were imaged using the aforementioned SEMs at similar accelerating voltages.

**As-printed chemical segregation characterization.** EPMA was performed using a Cameca model SX100 with five wavelength dispersive spectrometers and a CeB$_6$ thermionic emission gun. To quantify the amount of chemical segregation present after printing in both the EBM and SLM samples, a 20 × 20 grid scan of evenly spaced points with long dwell times (1 min of collection time per point) was collected in a 100 × 100 micron area on cross-sections taken 1 mm below the final build layer perpendicular to the build direction. Chemical maps with a reduced dwell time were used for visualization of this chemical segregation. Probe Image and Probe for EPMA software packages developed by Probe Software, Inc. were used for quantification of the collected spectra. All EPMA data underwent standards-based quantification with a combination of pure alloy standards and a reference alloy sample with composition Co–6.7W–8.9Al–3.3Cr–1.5Ta (at.%) that was previously characterized to high accuracy by inductively coupled plasma compositional analysis.

The collected EPMA grid scan data were sorted from highest Co concentration to lowest and assigned an apparent fraction solidified ($f_s$) from 0 to 1, as shown in Fig. 2d–f. It is assumed that the point with the highest Co content can be assigned an apparent $f_s = 0$, since Co partitions to the dendrite core during solidification, as

shown in Fig. 3a, resulting in the liquid becoming depleted of Co as solidification continues. Co was selected for sorting since it has the highest concentration in SB-CoNi-10 compared to the other alloying additions. In order to determine the apparent distribution coefficient, $k$, for each major alloying addition, the sorted concentration of the solid vs. apparent fraction solidified curves were fit using the Scheil equation[55]

$$C_s = kC_o(1 - f_s)^{k-1} \qquad (1)$$

where $C_s$ is the concentration of the solid, $C_o$ is the starting composition of the melt, $f_s$ is the fraction solidified, and the distribution coefficient is defined as $k = C_s/C_l$. Curve fits to the Scheil equation were made using a least-squares regression using data with apparent fraction solidified values between $f_s = 0.1$ and $f_s = 0.9$ for both the EBM and SLM datasets. These curve fits provided values for $C_o$ and $k$ using the values of $C_s$ and $f_s$ as inputs determined experimentally by the EPMA grid scan.

**Mechanical testing.** Sheet dogbones for mechanical testing were removed from the EBM-fabricated bars using electric discharge machining. The dogbones were 44 mm long with gauge dimensions of 2 mm × 3 mm × 14 mm. To remove the EDM-affected zone on the surface of the specimens, the gauge sections were ground down to 1200 grit using SiC grinding papers. Tensile tests on the EBM manufactured specimens were performed along the build direction (i.e. $Z$-orientation) at a strain rate of $1 \times 10^{-4}\,s^{-1}$ using an Instron 5582 universal testing machine. The SLM manufactured specimens were mechanically tested in the $Z$- and $XY$-orientation according to ASTM E8/E8M using machined cylindrical threaded dogbones with a gauge diameter of 6.35 mm. The strain rate was $3 \times 10^{-5}\,s^{-1}$ before yield and $1 \times 10^{-4}\,s^{-1}$ after yield. All mechanical tests were performed at room temperature and taken to rupture. EBSD maps of post-mortem specimens were collected after cross-sectioning the samples in half along the build direction.

## Data availability

The data that support the findings of this study are available from the corresponding author upon reasonable request.

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

## Acknowledgements

The authors gratefully acknowledge funding for this research provided by a Department of Defense (DoD) Vannevar Bush Faculty Fellowship, Grant ONR N00014-18-3031. In addition, tuition and stipend funding was provided to S.P.M. by a DoD National Defense Science and Engineering Graduate Fellowship. The research reported here made use of shared facilities of the National Science Foundation (NSF) Materials Research Science and Engineering Center (MRSEC) at UC Santa Barbara, DMR-172025. The UC Santa Barbara MRSEC is a member of the Materials Research Facilities Network (www.mrfn. org). A portion of this research was sponsored by the US Department of Energy, Office of Energy Efficiency and Renewable Energy, Advanced Manufacturing Office, under contract DE-AC05-00OR22725 with UT-Battelle, LLC and performed in partiality at the Oak Ridge National Laboratory's Manufacturing Demonstration Facility, an Office of Energy Efficiency and Renewable Energy user facility.

## Author contributions

N.Z., S.A.J.F., and W.E.S. were involved with fabrication of powder for both EBM and SLM prints and also carried out printing of the SLM material. P.N., M.M.K., and R.R.D. were involved with printing of EBM material and process development for the HIP treatment. C.J.T. performed electric discharge machining of the EBM mechanical test samples, and C.J.T., N.Z., and S.A.J.F. performed room temperature mechanical tests. G.G.E.S. collected EPMA chemical maps and grid scans on as-printed material. S.P.M., K.M.P., and A.T.P. characterized the microstructures and performed EBSD dataset collection. S.P.M. analyzed mechanical test data and chemical segregation data. S.P.M. drafted the initial manuscript. S.P.M., K.M.P., A.T.P., and T.M.P. edited and revised the initial manuscript. T.M.P. conceived, designed, and led the project.

## Competing interests

UCSB has a pending patent (T.M.P. and S.P.M. as inventors) on SB–CoNi alloys: "High Temperature Oxidation Resistant Co-based Gamma/Gamma Prime Alloys DMREF-Co", US patent application number US16/375,687, publication number US20200140978A1, international patent application number PCT/US2019/025882, international publication number WO2019195612A1. All other authors declare no competing interests.
