## [Peer Review File · Nature Communications]

REVIEWER COMMENTS

Reviewer #1 (Remarks to the Author):

The authors present a study of the processing via selective laser melting (SLM) and electron beam melting (EBM) of a Co-Ni alloy with composition Co-26.4Ni-13.2Al-6Cr-3.5Ta-1W (at%). The processing of Ni superalloys using additive manufacturing techniques presents nowadays some constraints owing to the formation of cracks during processing of the alloys as a result of different mechanisms. Strategies used in other alloy systems to mitigate crack formation during solidification and/or cooling are not applicable to Ni superalloys (e.g. control of grain nucleation can result in small grain sizes, which is detrimental for creep properties). It is in this context that the development of new high temperature alloys tailored to the metallurgical conditions of additive manufacturing is a timely matter that needs thorough research efforts. All these points are nicely addressed and introduced by the authors at the beginning of the manuscript.

While they even set interesting hypotheses that can guide the design of a crack-free superalloy for additive manufacturing ("control of the solvus through higher order alloying elements could delay the onset of gamma' precipitation after solidification" and further "the existence of a design space with a high resistance to cracking mediated by liquid films"), it is disappointing that they do not describe at all how they came to the proposed composition. Instead, they mention, in very general terms, that the composition was determined "by utilizing a suite of modern research tools, ..." and give a number of citations [27 - 30] that seem related to this but, as far as I can tell, are not particularly dedicated to additive manufacturing matters and, therefore, do not appear to be an integral part of the work presented in the manuscript. Why not describe how the new alloy was discovered? This may be one of the most relevant scientific achievements of this work.

Moreover, the rest of the manuscript gives the impression of a technical report instead of a scientific work that provides new insights to advance, e.g., the understanding of the mechanical behavior of the new alloy, its microstructure formation during SLM/EBM or the development of new high strength alloys, etc.

Besides these general considerations that should be addressed thoroughly before considering the manuscript for publication, there are some further specific points that need attention:

- The results shown in Fig. 2 and the corresponding text are very interesting but the authors indicate that the microstructures observed are a result of the solidification process (and they do their analysis of partition coefficients considering only the solidification stage). However, it is well known that during powder bed fusion the material is subjected to a series of sharp thermal cycles even in the solid state, i.e. after solidification. As a result, the material undergoes a sort of intrinsic heat treatment during manufacturing that may as well modify the solidified microstructure. Why wasn't this considered in the analysis of the as-built microstructures? The authors even show in Fig. 3 that the EBM microstructure changes as a function of depth (Fig. 4 a-d). At which height were the samples for EPMA analysis?

- Why did the authors study only RT tensile properties? The new alloy is supposed to be a candidate superalloy and, as such, its high temperature properties are much more relevant than tensile strength and ductility at room temperature. What's the relevance of RT investigations? How many samples were tested for each condition?

-Also, the comparison of the tensile test results is presented for samples tested at different orientations for different conditions (SLM materials were tested in XY direction, while the others in Z direction). This makes a direct comparison practically impossible. The authors should complement their investigations to have a representative picture of the behaviour of the alloy in the different thermal conditions and orientations.

Moreover, the interpretation of the tensile behavior of the alloy should contain scientific insights and not a mere comparison of the results obtained. There are asseverations that do not seem to provide any new insights into the mechanics of deformation of the material, e.g. "strain accumulates in grains oriented favorably for dislocation slip" (why shouldn't it?).

All in all, it's interesting (and very important) to know that a new alloy for high temperature applications may have been found/designed. However, the scientific novelty of the manuscript should be made clearer before this work can be considered for publication.

Reviewer #2 (Remarks to the Author):

The authors report on a novel class of high-temperature alloys to be employed for additive manufacturing (AM). Most alloys used in AM so far were not developed for the unique processing conditions prevailing in AM. Focusing on the well established powder bed techniques, i.e. electron beam melting (EBM) and selective laser melting (SLM), also referred to as laser powder bed fusion (LPBF), process induced cracking and further detrimental microstructural features are reported for many different alloy systems. A class of alloys known to be highly suffering from process induced cracking are Ni-base Superalloys. These alloys are employed in numerous demanding high-temperature applications. Even if data published in literature reveal that alloys such as CMSX-4 can be processed by EBM, none of the alloys available so far can be robustly processed in SLM and EBM. Mostly, processing by SLM suffers from the limited build chamber temperature, which is about 200 °C for most systems available on the market (even if novel systems are able to reach higher temperatures).

Based on their impressive experience in Co-base high-temperature alloys and use of cutting edge approaches for rapid alloy design (detailed in the first paragraph on page 4) the authors did a tremendous job. As is revealed by microstructure analysis and quasi-static tensile testing, the alloy developed is able to overcome present limitations of AM processed high-temperature alloys. Results shown and discussed are sound and conclusions drawn absolutely convincing.

However, a few aspects should be considered to further strengthen the conclusions of present work. Thus, the reviewer suggests revision of the paper. The following aspects should be addressed in the revised version of the manuscript:

In the introduction section worldwide activities towards design of novel alloys for AM should be mentioned (besides high-temperature materials) to further highlight the importance of this topic as well as the activities in this field.

For reference (and direct comparison to results presented) literature reporting on AM processing of Ni-base Superalloys, most importantly data reporting on EBM CMSX-4, have to be provided in higher number.

A rationale should be provided why analysis of chemical composition was conducted on the XY-plane and not on a plane parallel to build direction, e.g. XZ-plane.

As the paper reports on both EBM and SLM/LPBF processed conditions, microstructure evolution upon SLM (as-built and heat treated) has to be provided in the same kind data for the EBM condition were shown (Fig. 3). As is revealed by the tensile tests, yield strength of the SLM processed condition is higher as compared to the EBM material. Thus, significant contribution of the precipitates (and further microstructural features) seem to prevail. The contributing microstructural features have to be analysed in depth.

Moreover, for micrographs shown in Fig. 3 (g-j) the areas where these were taken from should be highlighted in the corresponding EBSD maps (Fig. 3 (e-f)).

A rationale for testing SLM and EBM processed materials in different orientations has to be provided. Testing of at least one of these conditions in a second orientation (i.e. SLM in z-direction

or EBM in horizontal direction) would significantly strengthen discussion and conclusions.

Fracture surfaces upon tensile testing should be analysed in more detail. The areas where Fig. 4 (d-e) were obtained from should be highlighted in Fig. 4 (b-c).

Results presented in Table 1 should include at least one set of data reporting on a SLM processed Ni-base Superalloy.

In the methods section further details have to be provided in terms of AM processing, e.g. the scan rotation between consecutive layers.

Were cooling rates during heat treatment of the SLM processed material the same as in case of the EBM material?

Appearance of Figure 4 (b-e) looks like these micrographs have been obtained by scanning electron microscopy, however, the methods section states that an optical microscope has been employed. Please double-check.

Finally, a rationale has to be provided, why tests for EBM and SLM material were conducted using different sample geometries.

In summary, the paper addresses a very important topic and, eventually, provides for answers allowing to tackle currently prevailing research gaps. In general, findings presented are of highest novelty and, thus, will significantly contribute to further advances in the field of new materials designed for AM. As AM in general will have a huge impact on future production in numerous branches, impact will not be limited to materials science. Moreover, the paper will be the basis for novel designs in industry sectors such as energy and mobility. Thus, number of citations is expected to be extremely high on the long run. As the results presented will be transferable to other alloy systems, the paper will significantly stimulate development of new materials for AM.

Reviewer #3 (Remarks to the Author):

This article describes a Co-Ni alloy designed for additive manufacture by a combination of trendy techniques well described in other articles. A heat treatment is devised and described and the alloy tested at room temperature for tensile properties both in the build direction and perpendicular to this. The alloys shows excellent ductility in both directions and is somewhat surprisingly stronger perpendicular to the growth direction, although this is not explained. The degree of micro-segregation is much lower for the SLM sample than the EBM material, attributed to the higher temperature gradient, but this is not related to the tensile performance.

The use of the Scheil equation to sort the grid of composition data demonstrates the much higher degree of homogeneity in the as solidified structure not only for this alloy but also from the SLM process. It is measuring the apparent partition coefficient, not the solid-liquid partitioning according to the assumptions of the Scheil equation. Hence the coefficients derived are a result not only of the fundamental partitioning, but also the very rapid cooling suppressing diffusion not only in the solid state but also in the liquid, and including any effect of repeated re-melting as the data is taken, as I understand, from near the top but not the top layer. An additional comment on this latter point would be useful.

The paper presents a promising alloy targeted at additive manufacturing and gives basic properties and characterisation. It gives a good overview of one of the fundamental problems with printing high strength alloys for high temperature applications – the difficulty in accommodating the thermal stresses inherent in the process in an alloy designed specifically to resist these. The paper does not give much new detail or insight into the design process itself, but does reference other suitable sources. It is not quite clear what the design criteria were apart from the rather generic

“favorable for 3D printing and that also possess favorable mechanical and environmental properties”. The alloy developed bears a notable similarity to that developed and described in reference 25 by one of the authors substituting Ta for Ti and increasing the Al content. The strategy of lowering the γ' solvus so that the formation of the precipitates is delayed allowing relaxation of the build stresses is stated, but the alloy is judged by the room temperature strength. A low γ' solvus is also likely to compromise the high temperature strength by reducing the amount of γ' at the higher temperatures. So, although it is very encouraging that the alloy shows such great ductility at RT, and has moderate strength, the proof of concept lies in the high temperature strength not being too compromised by the low γ' solvus, and indeed on any increases in alloy strength brought about by further optimisation not compromising the stress relaxation.

This is a very good paper, well written, rich in descriptive detail and offering a promising alloy for additive manufacturing. However, It lacks clarity in where the novelty is, whether in the methods used in the development of the alloy, the concept of a low γ' solvus for additive applications, in achieving an alloy that satisfies these aims, or indeed all three. For publication I would like to see the paper developing these specific issues and highlighting the relationships that lead to the improved performance. For instance; comparisons of the γ' solvus temperatures of the Alloy MarM247 with the new alloy, inclusion of some high temperature tensile properties to establish that the low solvus does not excessively damage the high temperature performance, or indeed that the very weakness of the alloy during manufacture is what allows it to retain ductility in service. With these additions I think it would certainly merit publication in Nature because of the importance of developing bespoke alloys which can best exploit this important new manufacturing route.

Enclosed is our response to the three reviewers who assessed our original manuscript. The comments of the reviewers are colored (Reviewer #1, Reviewer #2, and Reviewer #3) whereas our responses are in black.

Reviewer #1: The authors present a study of the processing via selective laser melting (SLM) and electron beam melting (EBM) of a Co-Ni alloy with composition Co-36.5Ni-13.2Al-6Cr-3.5Ta-1W (at%). The processing of Ni superalloys using additive manufacturing techniques presents nowadays some constraints owing to the formation of cracks during processing of the alloys as a result of different mechanisms. Strategies used in other alloy systems to mitigate crack formation during solidification and/or cooling are not applicable to Ni superalloys (e.g. control of grain nucleation can result in small grain sizes, which is detrimental for creep properties). It is in this context that the development of new high temperature alloys tailored to the metallurgical conditions of additive manufacturing is a timely matter that needs thorough research efforts. All these points are nicely addressed and introduced by the authors at the beginning of the manuscript.

We appreciate your careful reading of our manuscript.

Reviewer #1: While they even set interesting hypotheses that can guide the design of a crack-free superalloy for additive manufacturing (“control of the solvus through higher order alloying elements could delay the onset of gamma’ precipitation after solidification” and further “the existence of a design space with a high resistance to cracking mediated by liquid films”), it is disappointing that they do not describe at all how they came to the proposed composition. Instead, they mention, in very general terms, that the composition was determined “by utilizing a suite of modern research tools, ...” and give a number of citations [27 - 30] that seem related to this but, as far as I can tell, are not particularly dedicated to additive manufacturing matters and, therefore, do not appear to be an integral part of the work presented in the manuscript. Why not describe how the new alloy was discovered? This may be one of the most relevant scientific achievements of this work.

We agree that the design process for how this alloy composition was arrived at would be useful for the reader, so we have included a description of the tools and methods used for exploring this CoNi-base compositional space to arrive at the described alloy composition.

Reviewer #1: Moreover, the rest of the manuscript gives the impression of a technical report instead of a scientific work that provides new insights to advance, e.g., the understanding of the mechanical behavior of the new alloy, its microstructure formation during SLM/EBM or the development of new high strength alloys, etc.

We hope our extensions of these topics and other considerations in the Design Approach section, Results section, and Discussion section provide more of a sense of a scientific work by placing the collected data into a broader context in the literature rather than presenting data without the sufficient context and discussion. The original version of the paper was constrained in length and we are now pleased to be able to provide more details.

Reviewer #1: The results shown in Fig. 2 and the corresponding text are very interesting but the authors indicate that the microstructures observed are a result of the solidification process (and they do their analysis of partition coefficients considering only the solidification stage). However, it is well known that during powder bed fusion the material is subjected to a series of sharp thermal cycles even in the solid state, i.e. after solidification. As a result, the material undergoes a sort of intrinsic heat treatment during manufacturing that may as well modify the solidified microstructure. Why wasn’t this considered in the analysis of the as-built microstructures? The authors even show in Fig. 3 that the EBM microstructure changes as a function of depth (Fig. 4 (a-d)). At which height were the samples for EPMA analysis?

We agree that it is important to recognize that the additive process introduces a type of intrinsic heat treatment in as-printed microstructures, along with complications due to the elevated thermal gradients and cooling rates characteristic of the process. This detail has now been considered in the manuscript by comparing the collected EPMA compositional results on the EBM material to a single crystal alloy of SB-

CoNi-10 that was fabricated by the Bridgman technique. The intensity of the segregation is similar in the EBM material compared to the single crystal material, though the scale of the dendritic structure is clearly refined in the EBM material due to the higher cooling rates in the EBM process. While the layer-by-layer thermal excursions in the EBM process might be expected to influence the segregation to some degree due to subsequent diffusion, similarly, in a single crystal growth process there is an inherent "annealing" due to the slow withdrawal rates. Our prior research on segregation in these materials has shown these effects to be minor. These points have now been included in the discussion section. Also, in recognizing that the chemical measurements on the SLM material simply show that the segregation is too fine to measure by a SEM/EPMA based technique, we have removed the curve fits from the experimental Scheil curves in Fig. 2(f) for the SLM samples in order to not mislead or confuse the reader. Further experiments involving STEM-EDX mapping in the transmission electron microscope are in order to properly assess the segregation in as-printed SLM alloys.

Reviewer #1: Why did the authors study only RT tensile properties? The new alloy is supposed to be a candidate superalloy and, as such, its high temperature properties are much more relevant than tensile strength and ductility at room temperature. What's the relevance of RT investigations? How many samples were tested for each condition?

The main motivation for RT tensile testing were that cracking and ductility are a grave concern for high γ' volume fraction superalloys fabricated by AM. Our room temperature tensile tests are a proof of concept that the EBM and SLM alloys have been printed successfully without a high density of pre-existing cracks and flaws, which is evidenced by their mechanical performance even in the as-printed condition before HIP and heat treatment. A full industrial scale-up of this alloy would require additional testing of high temperature properties that must be considered for service, including but not limited to high temperature tensile properties, creep strength, and low and high cycle fatigue properties, in addition to well characterized oxidation response and coatability with conventional thermal barrier coatings. These could all be pursued in the future, once the proof of concept, as demonstrated by this paper, has been achieved. The stress-strain curves in the article represent all of the mechanical tests performed so far in the described heat-treatment conditions.

Reviewer #1: Also, the comparison of the tensile test results is presented for samples tested at different orientations for different conditions (SLM materials were tested in XY direction, while the others in Z direction). This makes a direct comparison practically impossible. The authors should complement their investigations to have a representative picture of the behaviour of the alloy in the different thermal conditions and orientations. Moreover, the interpretation of the tensile behavior of the alloy should contain scientific insights and not a mere comparison of the results obtained. There are asseverations that do not seem to provide any new insights into the mechanics of deformation of the material, e.g. "strain accumulates in grains oriented favorably for dislocation slip" (why shouldn't it?).

We agree that the direct comparison between the EBM and SLM materials is not fully justified due to the different orientations of the builds of these specimens. In order to improve upon this we have separated the EBM and SLM tensile test results into two separate stress-strain curves and have also collected additional tests on SLM material that was fabricated and tested along the Z-direction. Additional EBM powder was not available, so additional tests in the XY-direction for the EBM material has not been acquired, however the Z-direction built samples are likely the most technologically relevant since the microstructure resembles the microstructure of directionally solidified (DS) Ni-base superalloys. Additionally, discussions have been added in order to make the tensile results less of a comparison and more focused on the response of the alloys to different build orientations and thermal treatments before testing. Additionally, we have clarified that the EBSD of the post-mortem gauge sections were meant to illustrate that twinning induced plasticity (TWIP) or transformation induced plasticity (TRIP) effects were not exhibited by this alloy, that grain rotations are responsible for the elliptical fracture surfaces of the as-printed specimens, and that the excellent ductility of the alloy occurs in spite of the intense dislocation activity in grains favorably oriented for slip.

Reviewer #1: All in all, it's interesting (and very important) to know that a new alloy for high temperature applications may have been found/designed. However, the scientific novelty of the manuscript should be made clearer before this work can be considered for publication.

We appreciate your helpful suggestions, and we hope that the enclosed revisions to the article highlight the scientific novelty of our findings more clearly now.

Reviewer #2: The authors report on a novel class of high-temperature alloys to be employed for additive manufacturing (AM). Most alloys used in AM so far were not developed for the unique processing conditions prevailing in AM. Focusing on the well established powder bed techniques, i.e. electron beam melting (EBM) and selective laser melting (SLM), also referred to as laser powder bed fusion (LPBF), process induced cracking and further detrimental microstructural features are reported for many different alloy systems. A class of alloys known to be highly suffering from process induced cracking are Ni-base Superalloys. These alloys are employed in numerous demanding high-temperature applications. Even if data published in literature reveal that alloys such as CMSX-4 can be processed by EBM, none of the alloys available so far can be robustly processed in SLM and EBM. Mostly, processing by SLM suffers from the limited build chamber temperature, which is about 200 °C for most systems available on the market (even if novel systems are able to reach higher temperatures).

Based on their impressive experience in Co-base high-temperature alloys and use of cutting edge approaches for rapid alloy design (detailed in the first paragraph on page 4) the authors did a tremendous job. As is revealed by microstructure analysis and quasi-static tensile testing, the alloy developed is able to overcome present limitations of AM processed high-temperature alloys. Results shown and discussed are sound and conclusions drawn absolutely convincing.

However, a few aspects should be considered to further strengthen the conclusions of present work. Thus, the reviewer suggests revision of the paper. The following aspects should be addressed in the revised version of the manuscript:

We appreciate your careful reading of our manuscript, and we respond to your suggestions below.

Reviewer #2: In the introduction section worldwide activities towards design of novel alloys for AM should be mentioned (besides high-temperature materials) to further highlight the importance of this topic as well as the activities in this field. For reference (and direct comparison to results presented) literature reporting on AM processing of Ni-base Superalloys, most importantly data reporting on EBM CMSX-4, have to be provided in higher number.

More discussion on recent efforts in the literature to design novel alloys for AM have been added to the Introduction and Discussion sections in order to reflect the current global effort. These are references 23-27. While CMSX-4 was not developed specifically for additive manufacturing, there have been recent studies that have been able to fabricate single crystals of CMSX-4 through EBM by Körner et al. We have mentioned this work in the Discussion section in order to compare our results to theirs, with references to EBM CMSX-4 being Refs. 64 and 67.

Reviewer #2: A rationale should be provided why analysis of chemical composition was conducted on the XY-plane and not on a plane parallel to build direction, e.g. XZ-plane.

Measurements of the chemical composition were performed on the XY-plane in order to collect compositional data perpendicular to the dendrite growth direction, allowing for us to assign an apparent solid fraction to each data point with higher confidence. The interaction volume of the SEM spot with the alloy surface is significant at these length scales, therefore collection of chemical information on the XZ-plane could lead to confusion in the collected data due to possible compositional variations beneath the alloy surface. This has been made more clear in the manuscript text in the Results section.

Reviewer #2: As the paper reports on both EBM and SLM/LPBF processed conditions, microstructure evolution upon SLM (as-built and heat treated) has to be provided in the same kind data for the EBM condition were shown (Fig. 3). As is revealed by the tensile tests, yield strength of the SLM processed

condition is higher as compared to the EBM material. Thus, significant contribution of the precipitates (and further microstructural features) seem to prevail. The contributing microstructural features have to be analysed in depth.

We agree that the microstructure of the SLM printed material needed to be described in greater detail. To accomplish this, a new figure focused on the SLM microstructure in the as-printed and post-processed conditions has been included as Fig. 5. We have made this figure using similar sized EBSD scans and similar magnification SEM images as the EBM microstructure figure (Fig. 4) in order for the reader to more directly compare the EBM and SLM materials. Also, this figure made more clear that segregations on the order of the cellular structure are present in the SLM material, which may not be apparent from the chemical measurements and maps provided earlier in the text. The higher strengths and reduced ductilities of the SLM material are likely due to the difference in loading direction (XZ-direction vs. along build direction) and smaller grain sizes after heat treatment.

Reviewer #2: Moreover, for micrographs shown in Fig. 3 (g-j) the areas where these were taken from should be highlighted in the corresponding EBSD maps (Fig. 3 (e-f)).

These locations in what is now Fig. 4 have been indicated in the revised version, with a similar treatment for the new Fig. 5.

Reviewer #2: A rationale for testing SLM and EBM processed materials in different orientations has to be provided. Testing of at least one of these conditions in a second orientation (i.e. SLM in z-direction or EBM in horizontal direction) would significantly strengthen discussion and conclusions.

A rationale for why the EBM material was tested in the Z-orientation was provided in the discussion, since the columnar grain structure formed by EBM is similar to the industrially relevant columnar microstructure made through directionally solidified (DS) castings of superalloys. In order to strengthen our findings, and additional set of tensile tests has been performed on the SLM material in the Z-direction for both as-printed and fully post-processed samples, which has been included in Fig. 6 and compared to the horizontally built (XY-direction) SLM material.

Reviewer #2: Fracture surfaces upon tensile testing should be analysed in more detail. The areas where Fig. 4 (d-e) were obtained from should be highlighted in Fig. 4 (b-c).

More detail on why the observed fracture surfaces are indicative of ductile fracture has been added to paper. Additionally, since the SEM micrographs of these regions are very tiny compared to scale of the entire fracture surface, we have clarified in the Fig. 6 caption that the higher magnification SEM images are taken from near the center of each fracture surface, but have chosen not to indicate the exact locations for improved clarity of the images.

Reviewer #2: Results presented in Table 1 should include at least one set of data reporting on a SLM processed Ni-base Superalloy.

We agree that a comparison for the SLM material is needed, and so a comparison to a recent study on IN738LC, a similar NI-based high γ' -volume fraction superalloy, has been provided in both what is now Table 2 and in the mechanical properties figure, Fig. 6.

Reviewer #2: In the methods section further details have to be provided in terms of AM processing, e.g. the scan rotation between consecutive layers. Were cooling rates during heat treatment of the SLM processed material the same as in case of the EBM material?

In the Methods section, we have clarified that the EBM materials were fabricated using the standard raster scan strategy on the Arcam Q10+ system running 4.2.89 EBM control software. The scan rotation between consecutive layers is complex, so we feel this is the most accurate way to describe the scan strategy

to the reader. This is in contrast to the SLM materials that were fabricated using a bidirectional raster scan strategy (i.e. 90° rotation in the laser scan path between consecutive layer). Additional details on the heat treatment procedure have been included with regard to whether samples were oil quenched or furnace quenched after each step of the heat treatment process.

Reviewer #2: Appearance of Figure 4 (b-e) looks like these micrographs have been obtained by scanning electron microscopy, however, the methods section states that an optical microscope has been employed. Please double-check.

Correct, these micrographs were SEM micrographs, which has been clarified in the Methods section.

Reviewer #2: Finally, a rationale has to be provided, why tests for EBM and SLM material were conducted using different sample geometries.

The EBM samples were tested by the academic team at UC-Santa Barbara and the SLM samples were tested by the industrial team at Carpenter Technology Corporation. Each facility had the testing capability for the mechanical test geometries used, allowing work to continue in parallel during data collection. Both sample geometries can be considered large with many grains present across the sample gauge lengths, however the tests performed by the UC-Santa Barbara team are not the same as ASTM E8 and are only intended to describe the basic mechanical properties such as yield strength, ultimate tensile strength, and ductility. In the case of industrial scale-up, following ASTM standards would be desirable, which motivates the use of these standards by the industrial team at Carpenter Technology Corporation.

Reviewer #2: In summary, the paper addresses a very important topic and, eventually, provides for answers allowing to tackle currently prevailing research gaps. In general, findings presented are of highest novelty and, thus, will significantly contribute to further advances in the field of new materials designed for AM. As AM in general will have a huge impact on future production in numerous branches, impact will not be limited to materials science. Moreover, the paper will be the basis for novel designs in industry sectors such as energy and mobility. Thus, number of citations is expected to be extremely high on the long run. As the results presented will be transferable to other alloy systems, the paper will significantly stimulate development of new materials for AM.

We appreciate your helpful suggestions, and hope the changes made to the manuscript are satisfactory for acceptance.

Reviewer #3: This article describes a Co-Ni alloy designed for additive manufacture by a combination of trendy techniques well described in other articles. A heat treatment is devised and described and the alloy tested at room temperature for tensile properties both in the build direction and perpendicular to this. The alloys shows excellent ductility in both directions and is somewhat surprisingly stronger perpendicular to the growth direction, although this is not explained. The degree of micro-segregation is much lower for the SLM sample than the EBM material, attributed to the higher temperature gradient, but this is not related to the tensile performance.

Reviewer #3: The alloys shows excellent ductility in both directions and is somewhat surprisingly stronger perpendicular to the growth direction, although this is not explained.

A brief discussion has been provided explaining that the mechanical properties of additively manufacturing metals tends to vary with the loading direction, especially with long columnar-grained specimens. Additionally, we have separated the EBM and SLM results onto separate stress-strain curves as to avoid direct comparison, and have acquired additional mechanical testing results for the SLM samples parallel to the build direction to complement the tests that were performed perpendicular to the build direction.

Reviewer #3: The use of the Scheil equation to sort the grid of composition data demonstrates the much higher degree of homogeneity in the as solidified structure not only for this alloy but also from the SLM

process. It is measuring the apparent partition coefficient, not the solid-liquid partitioning according to the assumptions of the Scheil equation. Hence the coefficients derived are a result not only of the fundamental partitioning, but also the very rapid cooling suppressing diffusion not only in the solid state but also in the liquid, and including any effect of repeated re-melting as the data is taken, as I understand, from near the top but not the top layer. An additional comment on this latter point would be useful.

Throughout the text we have indicated that the experimentally measured distribution coefficients are indeed the “apparent” distribution coefficients, due to the factors described by the reviewer. Additional comments on the effect of thermal gradient have been included in the discussion. Additionally, a data set on a Bridgman grown single crystal of SB-CoNi-10 has been included for comparison in Fig. 2(a), and the measured apparent distribution coefficients are compared in the new Table 1. The Bridgman growth process has much smaller thermal gradients, and also isn’t subject to the effects of re-melting or preheating in the way the EBM and SLM processes are, and yet the distribution coefficients are similar between Bridgman-grown SB-CoNi-10 and EBM processed SB-CoNi-10. This is also highlighted by the similarities in the Scheil curves displayed in Fig. 2(d) and 2(e).

Reviewer #3: The paper presents a promising alloy targeted at additive manufacturing and gives basic properties and characterisation. It gives a good overview of one of the fundamental problems with printing high strength alloys for high temperature applications – the difficulty in accommodating the thermal stresses inherent in the process in an alloy designed specifically to resist these. The paper does not give much new detail or insight into the design process itself, but does reference other suitable sources. It is not quite clear what the design criteria were apart from the rather generic “favorable for 3D printing and that also possess favorable mechanical and environmental properties”. The alloy developed bears a notable similarity to that developed and described in reference 25 by one of the authors substituting Ta for Ti and increasing the Al content.

In order to describe the tools used in the design process more effectively, we have added a "Design Methods" section to the manuscript. Particularly useful was the combinatorial synthesis technique that allowed for exploration of the complex compositional space needed to discover alloys with high γ' -solvus, two-phase microstructures, and good oxidation resistance. Previous understanding of the solidification behavior of this alloy through single crystal casting with the Bridgman method informed decisions to process this alloy through EBM and SLM for the above study. We acknowledge that that are a few examples in the literature of CoNi-based superalloys that possess somewhat similar alloy compositions, since the community has found that additions of Ni, Cr, Ta, Ti, and other additions are needed to develop alloys with both high temperature stability and oxidation resistance. Differences of only a few atomic percent of certain alloying additions can have significant effects on the alloy microstructure and properties. For example, the alloy described in what was Ref. 25 (Forsik, S. A. J. et al. High-temperature oxidation behavior of a novel Co-base superalloy. *Metall. Mater. Trans. A* **49**, 4058–4069 (2018)) has 5 similar majority elements to the investigated SB-CoNi-10 in our work, but these alloying additions are in very different proportions. This results in their case with alloy with a reduced γ' volume fraction, a reduced γ' solvus by over 100 °C, and excellent oxidation resistance. Depending on the alloy application or intended processing pathway (for example, this alloy was processed through a forging pathway), this combination of properties may be preferable over what we have described with SB-CoNi-10. Since our recent research, along with that of other researchers, has discovered balanced alloy properties in these CoNi-based compositional spaces, there seems to be future opportunities for tuning alloy properties by modifying the global alloy compositions with the intended processing pathway of additive manufacturing in mind.

Reviewer #3: The strategy of lowering the γ' solvus so that the formation of the precipitates is delayed allowing relaxation of the build stresses is stated, but the alloy is judged by the room temperature strength. A low γ' solvus is also likely to compromise the high temperature strength by reducing the amount of γ' at the higher temperatures. So, although it is very encouraging that the alloy shows such great ductility at RT, and has moderate strength, the proof of concept lies in the high temperature strength not being too compromised by the low γ' solvus, and indeed on any increases in alloy strength brought about by further optimisation not compromising the stress relaxation.

The use of room temperature tensile testing was primarily used to demonstrate that the alloy had good mechanical integrity in the as-printed and heat-treated conditions, since defects such as a high density of pre-existing cracks or flaws are especially prevalent in precipitation strengthened superalloys fabricated by additive manufacturing. We agree that the high temperature mechanical properties would be desirable in future study, and would need to be investigated if such an alloy was to be introduced into a high-temperature, load-bearing application, such as blading for the hot sections of turbine engines. With respect to the concern that the high temperature strength may be compromised by the slightly reduced γ' solvus, it is important to recognize that the γ' solvus in the described SB-CoNi-10 alloy is still high, with a value near 1200 °C. A peak in the flow stress occurs in the range of 600-800 °C for Ni-, CoNi- and Co-based superalloys alike, above which a reduction in strength occurs. Aging studies on the SB-CoNi-10 alloy at various temperatures and aging times has revealed that the high γ' volume fraction is retained up until roughly 1050 °C, after which the volume fraction reduces until the γ' solvus is reached. Therefore, the γ' solvus likely doesn't affect the strength as much as other factors such as the reduced planar fault energies (e.g. $\gamma_{APB,111}$) that may be present in CoNi-based alloy compositions, as indicated by the high temperature creep studies on single crystals of various CoNi-based superalloys by Eggeler et al. (Eggeler, Y. M., Titus, M. S., Suzuki, A., & Pollock T. M. *Acta Mater.* **77**, 352–359 (2014)) and Titus et al. (Titus, M. S., Eggeler, Y. M., Suzuki, S., & Pollock, T. M. *Acta Mater.* **82**, 530–539 (2015))

Reviewer #3: This is a very good paper, well written, rich in descriptive detail and offering a promising alloy for additive manufacturing. However, It lacks clarity in where the novelty is, whether in the methods used in the development of the alloy, the concept of a low γ' solvus for additive applications, in achieving an alloy that satisfies these aims, or indeed all three. For publication I would like to see the paper developing these specific issues and highlighting the relationships that lead to the improved performance. For instance; comparisons of the γ' solvus temperatures of the Alloy MarM247 with the new alloy, inclusion of some high temperature tensile properties to establish that the low solvus does not excessively damage the high temperature performance, or indeed that the very weakness of the alloy during manufacture is what allows it to retain ductility in service.

We appreciate your careful reading of our manuscript and the suggested improvements and edits. We hope the highlighted changes provide a clearer focus on how the investigated alloy was designed and also places the alloy in a broader context compared to similar Ni-base superalloys manufactured by both SLM and EBM.

REVIEWERS' COMMENTS:

As I mentioned above, Reviewer #1 was not available to review this time.

Reviewer #2 (Remarks to the Author):

The authors addressed all issues raised by all reviewers thoroughly. Changes made strengthened the discussion and the conclusion sections considerably. Design strategies detailed can be transferred to other alloy systems. Thus, the impact of the current work will be very high, such that the work is absolutely to the standards of Nature communications. I don't see any open issues remaining and, thus, recommend acceptance of the current version of the manuscript.

As I already highlighted the importance of the current work in my initial review, I copied the most important parts:

The authors report on a novel class of high-temperature alloys to be employed for additive manufacturing (AM). Most alloys used in AM so far were not developed for the unique processing conditions prevailing in AM. Focusing on the well-established powder bed techniques, i.e. electron beam melting (EBM) and selective laser melting (SLM), also referred to as laser powder bed fusion (LPBF), process induced cracking and further detrimental microstructural features are reported for many different alloy systems. A class of alloys known to be highly suffering from process induced cracking are Ni-base superalloys. These alloys are employed in numerous demanding high-temperature applications. Even if data published in literature reveal that alloys such as CMSX-4 can be processed by EBM, none of the alloys available so far can be robustly processed in SLM and EBM. Mostly, processing by SLM suffers from the limited build chamber temperature, which is about 200 °C for most systems available on the market (even if novel systems are able to reach higher temperatures). Based on their impressive experience in Co-base high-temperature alloys and use of cutting edge approaches for rapid alloy design the authors did a tremendous job. As is revealed by microstructure analysis and quasi-static tensile testing, the alloy developed is able to overcome present limitations of AM processed high-temperature alloys. Results shown and discussed are sound and conclusions drawn absolutely convincing.

In summary, the paper addresses a very important topic and, eventually, provides for answers allowing to tackle currently prevailing research gaps. In general, findings presented are of highest novelty and, thus, will significantly contribute to further advances in the field of new materials designed for AM. As AM in general will have a huge impact on future production in numerous branches, impact will not be limited to materials science. Moreover, the paper will be the basis for novel designs in industry sectors such as energy and mobility. Thus, number of citations is expected to be very high on the long run. As the results presented will be transferable to other alloy systems, the paper will significantly stimulate development of new materials for AM. The thorough experimental effort, quality of data, in-depth discussion and expected impact of the approach presented are clearly up to the standard of Nature Communications.

In consequence, the reviewer strongly recommends acceptance of the current work in its revised version.

Reviewer #3 (Remarks to the Author):

The paper is very much improved incorporating the diverse comments from all three reviewers very skilfully and succinctly. The points about the development of the alloys are explored at length and the general descriptions of each technique are useful but could, if length is a problem, be omitted. The new testing also shows clearly the difference between the directions tested and by

presenting the tests in two graphs, clearly distinguishes between the techniques.

This is an excellent paper describing a systematic approach to alloy design incorporating physical modelling with extrapolation techniques to great effect. The alloy developed shows much improved ductility and strength and the points made about the potential high temperature properties are well made. The prospects for further development of these alloys to give much improved printable high temperature components are excellent.